# Racial disparities in post-transplant stroke and mortality following stroke in adult cardiac transplant recipients in the United States

**Lathan Liou**[1,2], **Elizabeth Mostofsky**[1], **Laura Lehman**[1,3,4], **Soziema Salia**[1,5], **Suruchi Gupta**[1,3,6], **Francisco J. Barrera**[1], **Murray A. Mittleman**[1,3,7]*

1 Department of Epidemiology, Harvard T.H. Chan School of Public Health, Boston, Massachusetts, United States of America, 2 Merck & Co., Merck Research Laboratories, Boston, Massachusetts, United States of America, 3 Harvard Medical School, Boston, Massachusetts, United States of America, 4 Department of Neurology, Boston Children's Hospital, Boston, Massachusetts, United States of America, 5 Department of Internal Medicine, Cape Coast Teaching Hospital, Cape Coast, Ghana, 6 Division of Endocrinology, Diabetes, and Metabolism, Department of Medicine, Beth Israel Deaconess Medical Center, Boston, Massachusetts, United States of America, 7 Division of Cardiovascular Medicine, Department of Medicine, Beth Israel Deaconess Medical Center, Boston, Massachusetts, United States of America

* mmittlem@hsph.harvard.edu

**Data Availability Statement:** The data are available from SRTR upon request by emailing srtr@srtr.org (More information can be found at https://www.srtr.org/requesting-srtr-data/data-requests/). SRTR

## Abstract

Black heart transplant recipients have a higher mortality rate than white recipients 6–12 months after transplant. Whether there are racial disparities in post-transplant stroke incidence and all-cause mortality following post-transplant stroke among cardiac transplant recipients is unknown. Using a nationwide transplant registry, we assessed the association between race and incident post-transplant stroke using logistic regression and the association between race and mortality among adults who survived a post-transplant stroke using Cox proportional hazards regression. We found no evidence of an association between race and the odds of post-transplant stroke (OR = 1.00, 95% CI: 0.83–1.20). The median survival time of those with a post-transplant stroke in this cohort was 4.1 years (95% CI: 3.0, 5.4). There were 726 deaths among the 1139 patients with post-transplant stroke, including 127 deaths among 203 Black patients and 599 deaths among 936 white patients. Among post-transplant stroke survivors, Black transplant recipients experienced a 23% higher rate of mortality compared to white recipients (HR = 1.23, 95% CI: 1.00–1.52). This disparity is strongest in the period beyond the first 6 months and appears to be mediated by differences in the post-transplant setting of care between Black and white patients. The racial disparity in mortality outcomes was not evident in the past decade. The improved survival of Black patients in the recent decade may reflect overall protocol improvements for heart transplant recipients irrespective of race, such as advancements in surgical techniques and immediate postoperative care as well as increased awareness about reducing racial disparities.

is a third-party organization that owns the data, which is why we are not allowed to share data per the Data Use Agreement that we signed with SRTR.

**Funding:** EM is funded by Grant # K01AA027831 from the National Institute on Alcohol Abuse and Alcoholism (NIAAA). The content is solely the responsibility of the authors and does not necessarily represent the official views of the funding organizations. The funders had no role in study design, data collection and analysis, decision to publish, or preparation of the manuscript.

**Competing interests:** The authors have declared that no competing interests exist.

**Abbreviations:** BMI, body mass index; CI, confidence interval; ECMO, extracorporeal membrane oxygenation; HR, hazards ratio; ICU, intensive care unit; OPTN, Organ Procurement and Transplantation Network; OR, odds ratio; SRTR, Scientific Registry of Transplant Recipients; VAD, ventricular assist device.

# Introduction

Heart transplant is the most effective treatment for advanced heart failure, increasing survival and improving quality of life [1]. In the past 5 years, the number of total heart transplants performed annually in the United States has been increasing [2]. However, heart transplantation carries risks such as stroke. Although the annual incidence of ischemic (1.7%) and hemorrhagic stroke (0.7%) is lower among heart transplant recipients compared to that of those awaiting transplantation (ischemic: 6.8%, hemorrhagic: 2.6%) [3], there is a higher mortality rate among those with a post-transplant stroke, especially beyond the first six months following the transplant [4].

In the general population, Black adults have a 3–4 times higher risk of dying from stroke than white adults [5]. Furthermore, among heart transplant recipients, Black adults have a higher mortality rate [6] than patients from other racial/ethnic groups, particularly in the period beyond 6 months following transplant [4]. Although mortality rates among transplant recipients have declined since 1987, the higher mortality rate among Black recipients has persisted [7]. However, whether there are racial disparities in post-transplant stroke incidence and/or long-term mortality following post-transplant stroke remains unknown. Therefore, we conducted a cohort study to examine whether there are disparities in post-transplant stroke incidence and/or long-term prognosis among heart transplant patients experiencing a post-transplant stroke using national data on cardiac transplant recipients.

# Methods

## Data source

We used data from the Scientific Registry of Transplant Recipients (SRTR), a national database that includes data on all donors, wait-listed candidates, and transplant recipients in the U.S. The SRTR contains information on deaths via linkage to the Social Security Master Death File. As described elsewhere [8], members of the Organ Procurement and Transplantation Network (OPTN) submit the information, and the Health Resources and Services Administration, U.S. Department of Health and Human Services, provides oversight. This study was reviewed and informed consent was waived by the Institutional Review Board at Harvard T.H. Chan School of Public Health due to the retrospective and anonymized nature of this registry dataset.

## Study population

This study includes 54,995 Black and white adult (>18 years) heart-only transplant recipients who received their first transplant between January 1, 1987, and December 31, 2018 with at least one recorded follow-up visit. The dataset we had available had follow-up until September 13, 2021. 5,838 adults of other reported racial and ethnic backgrounds were not included in this study cohort in order to focus the scope of this research study extending previous literature investigating disparities in outcomes between Black and white transplant recipients. We used the OPTN classifications of race, and we examined potential differences in disparities over time by defining three 10-year intervals: 1987–1999, 2000–2009, and 2010–2018 *a priori*.

**Statistical analysis.** We use the National Institute on Minority Health and Health Disparities (NIMHD) research framework and prior literature to guide our selection of mediating factors and interpretation of the association between race and outcome [9].

**Post-transplant stroke.** We defined post-transplant stroke as a stroke occurring within 30 days after the transplant procedure. We used conditional logistic regression stratified by transplant center to calculate odds ratios (OR) and 95% confidence intervals (95% CI) for the association between race and the incidence of post-transplant stroke. We adjusted for clinical

and demographic factors identified by previous studies examining the association between race and mortality [10, 11]. These factors are age (continuous), sex, body mass index (BMI) [underweight (<18 kg/m2), normal (18–24.9 kg/m2), overweight (25–29.9 kg/m2), obese (>30 kg/m$^2$)], cancer malignancy, diabetes, hypertension, etiology of heart disease (ischemic, non-ischemic dilated cardiomyopathy, other), prior stroke, history of cerebrovascular disease, smoking status (none, past/current, unknown), any prior ventricular assist device (VAD) use, extracorporeal membrane oxygenation (ECMO), life support use (none, life support device, life support without device, unknown), functional status (moribund/hospitalized/severely disabled, requires significant assistance, normal with minor/no symptoms), hospitalization status (intensive care unit [ICU], non-ICU hospitalization, not hospitalized), total ischemic time (minutes), pulmonary arterial systolic blood pressure (mm/Hg), creatinine (mg/dL), bilirubin (mg/dL), number of HLA-AB mismatches, number of HLA-DR mismatches, panel reactive antibodies (PRA) above 10%, acute transplant rejection, waitlist time (days) and transplant era (categorical).

In secondary analyses, we further adjusted for education level (below high school, high school, college and above) and insurance type (Medicare, Medicaid, private, other) to explore whether socioeconomic factors were driving associations between race and outcome.

**Post-transplant mortality.** We examined whether race was associated with post-transplant mortality among recipients who survived a post-transplant stroke. For this analysis, we used all person-time from the date of transplant until the date of death or administrative censoring on December 31, 2018. We also further adjusted for available post-stroke management covariates: any inotropic medication use (categorical), and setting of follow-up care (transplant center, non-transplant center, primary care physician [PCP], and other). We excluded recipients with no data on body mass index (1 Black patient and 12 white patients). We performed multiple imputation on only covariate data using chained equations (MICE) [12] using the R package *mice* with default settings [13]. We included our model covariates to impute missing values for categorical variables that contained missing values: BMI (1.8%), malignancy (22%), hypertension (39%), smoking status (32%), VAD (31%), life support status (0.2%), functional status (41%), hospitalization status (0.2%), PRA level (9%), education level (34%), insurance type (19%), history of cerebrovascular disease (22%) and setting of follow up care (54%) and for continuous variables that contained missing values: ischemic time (4%), systolic blood pressure (24%), creatinine (19%), bilirubin (22%), number of HLA-AB mismatches (23%), and number of HLA-DR mismatches (23%). We found that data were missing at random based on race and used the MAR assumption for multiple imputation [14]. We did not conduct complete case analyses as a sensitivity analysis due to a substantial lack of power.

We used Cox proportional hazards models to calculate hazard ratios (HR) and 95% confidence intervals (95% CI) for the association between race and all-cause mortality among post-transplant stroke survivors adjusting for the covariates listed above. We tested the proportional hazards assumption of the fully adjusted Cox model across the full range of follow-up using the "cox.zph" function from the R *survival* package.

Based on previous literature showing potential non-proportional hazards 6 months post-transplant [15], we examined the association between race and post-stroke mortality within 6 months of the transplant and beyond 6 months following the transplant. We first plotted Kaplan-Meier survival curves for both time windows, and we performed log-rank tests. For the analysis of the first 6 months, we set the index date as the date of discharge and followed patients until the time of death, last known follow-up, or 6 months, whichever came first. For the analysis beyond 6 months, we set the index date as 6-months following their date of discharge and a patient's status as dead, alive, or censored at the time of last known follow-up or

December 31, 2018, whichever came first. We then fit fully adjusted Cox models for both time-frames and tested the proportional hazards assumption for each of these models.

To examine whether socioeconomic factors at least partially drive the association between race and post-transplant mortality, we constructed fully adjusted models using the covariates listed above and education level and insurance. As a sensitivity analysis, we explored models not adjusting for the setting of follow-up care as this may be influenced by race/ethnicity and therefore may be a potential mediator rather than a confounding factor. We also explored whether any association between race and post-stroke mortality differed across *a priori* defined 10-year transplant era intervals both within 6 months and beyond 6 months. As an additional sensitivity analysis, we included transplant center in the multiple imputation process and ran all the Cox models including transplant center as a stratification variable.

We performed all analyses using R, version 4.1.0, for macOS. All hypothesis tests were 2-sided, with statistical significance defined as $p < 0.05$.

## Results

Among 54,995 white and Black adult patients who received their first heart transplant in the United States between 1987 and 2018, 1,139 (2.1%) experienced a post-transplant stroke, among whom 726 died over follow-up. Baseline characteristics and clinical features for the primary analysis cohort are summarized in Table 1. Of those who experienced a post-transplant stroke, 203 were Black, and 936 were white. Black recipients were, on average, younger than white recipients. Black recipients had a higher prevalence of non-ischemic cardiomyopathy (68% vs. 34%), they were more likely to have a severely disabled functional status (41% vs. 32%), and they were more likely to be on life support (91% vs. 81%). Baseline characteristics for Black and white post-transplant survivors across the three transplantation eras are summarized in Table 2. Over time, there is a tendency of a greater proportion of Black females and a greater proportion of Black recipients who have private insurance, obesity, hypertension, and high PRA values receiving heart transplants.

There was no difference in the incidence of post-transplant stroke between Black (2.1%) and white (2.1%) recipients, even after accounting for covariates (adjusted OR = 1.00, 95% CI: 0.83–1.20).

Among the 1,139 patients with post-transplant stroke, the median survival time was 4.1 years (95% CI: 3.0, 5.4), and there was a total of 726 deaths, including 127 deaths among 203 Black patients and 599 deaths among 936 white patients. Over the full duration of the study, Black post-transplant stroke survivors had a 23% higher rate of mortality compared to their white counterparts after adjusting for covariates (HR = 1.23, 95% CI: 1.00–1.52; Table 3, Model 1). Further adjustment for education and insurance did not meaningfully change the association (HR = 1.21, 95% CI: 0.97–1.50; Model 2). Additionally adjusting for the setting of post-transplant care again did not substantially change the association (HR = 1.19, 95% CI: 0.96–1.48; Model 3).

As expected, the global Schoenfeld residuals test showed evidence of a violation of proportional hazards in the fully adjusted model ($p < 0.01$). *A priori* based on prior literature, we examined the association between race and post-stroke mortality separately for two time-frames: within 6 months and beyond 6 months. Global Schoenfeld residuals tests showed no violation of proportional hazards in the fully adjusted model within 6 months ($p = 0.12$) and beyond 6 months ($p = 0.40$). The Kaplan-Meier curves suggest that there were no apparent racial differences in mortality within 6 months of post-transplant stroke (log-rank $p = 0.98$). However, among those who survived the first 6 months, white post-transplant stroke survivors had a higher survival probability than Black post-transplant stroke survivors (log-rank

**Table 1. Heart transplant recipient characteristics by race and post-transplant stroke, Scientific Registry of Transplant Recipients (SRTR), January 1, 1987 to December 31, 2018; n (%) unless otherwise specified.**

| | No Post-transplant Stroke | | Post-transplant Stroke | |
|---|---|---|---|---|
| | White (N = 44264) | Black (N = 9592) | White (N = 936) | Black (N = 203) |
| **Recipient Age** | | | | |
| Mean (SD) | 53.8 (11.4) | 49.0 (12.6) | 55.4 (10.7) | 50.7 (12.9) |
| Median [Min, Max] | 56.1 [18.0, 78.6] | 51.0 [18.0, 76.4] | 57.4 [18.5, 79.0] | 52.1 [18.7, 78.4] |
| **Recipient Sex** | | | | |
| Male | 34947 (79.0%) | 6354 (66.2%) | 701 (74.9%) | 132 (65.0%) |
| Female | 9317 (21.0%) | 3238 (33.8%) | 235 (25.1%) | 71 (35.0%) |
| **Education Level** | | | | |
| Below high school | 794 (1.8%) | 234 (2.4%) | 27 (2.9%) | 8 (3.9%) |
| High school | 11727 (26.5%) | 3613 (37.7%) | 325 (34.7%) | 79 (38.9%) |
| College and above | 15601 (35.2%) | 3504 (36.5%) | 430 (45.9%) | 89 (43.8%) |
| **Insurance** | | | | |
| Medicare | 2726 (6.2%) | 1659 (17.3%) | 64 (6.8%) | 38 (18.7%) |
| Medicaid | 9302 (21.0%) | 2513 (26.2%) | 258 (27.6%) | 73 (36.0%) |
| Private | 21110 (47.7%) | 3849 (40.1%) | 555 (59.3%) | 87 (42.9%) |
| Other | 1560 (3.5%) | 494 (5.2%) | 41 (4.4%) | 5 (2.5%) |
| **BMI Categories** | | | | |
| Underweight | 797 (1.8%) | 228 (2.4%) | 12 (1.3%) | 6 (3.0%) |
| Normal or health weight | 16797 (37.9%) | 3449 (36.0%) | 348 (37.2%) | 70 (34.5%) |
| Overweight | 16337 (36.9%) | 3227 (33.6%) | 346 (37.0%) | 71 (35.0%) |
| Obese | 9492 (21.4%) | 2565 (26.7%) | 218 (23.3%) | 55 (27.1%) |
| **Malignancy** | | | | |
| No | 31119 (70.3%) | 7886 (82.2%) | 825 (88.1%) | 192 (94.6%) |
| Yes | 2166 (4.9%) | 456 (4.8%) | 70 (7.5%) | 9 (4.4%) |
| **Smoker** | | | | |
| No | 12299 (27.8%) | 4156 (43.3%) | 310 (33.1%) | 109 (53.7%) |
| Yes | 16624 (37.6%) | 3409 (35.5%) | 471 (50.3%) | 75 (36.9%) |
| **Hypertension** | | | | |
| No | 15424 (34.8%) | 2954 (30.8%) | 370 (39.5%) | 61 (30.0%) |
| Yes | 11239 (25.4%) | 3155 (32.9%) | 322 (34.4%) | 82 (40.4%) |
| **Diabetes Mellitus** | | | | |
| No | 36570 (82.6%) | 7497 (78.2%) | 696 (74.4%) | 139 (68.5%) |
| Yes | 7694 (17.4%) | 2095 (21.8%) | 240 (25.6%) | 64 (31.5%) |
| **Prior Stroke** | | | | |
| No | 42189 (95.3%) | 8972 (93.5%) | 800 (85.5%) | 170 (83.7%) |
| Yes | 2075 (4.7%) | 620 (6.5%) | 136 (14.5%) | 33 (16.3%) |
| **History of Cerebrovascular Disease** | | | | |
| No | 31868 (72.0%) | 7856 (81.9%) | 835 (89.2%) | 177 (87.2%) |
| Yes | 1496 (3.4%) | 488 (5.1%) | 62 (6.6%) | 21 (10.3%) |
| **Cardiovascular Etiology** | | | | |
| Ischemic | 22291 (50.4%) | 2112 (22.0%) | 517 (55.2%) | 47 (23.2%) |
| Non-ischemic | 17527 (39.6%) | 6776 (70.6%) | 317 (33.9%) | 138 (68.0%) |
| Other | 4442 (10.0%) | 704 (7.3%) | 102 (10.9%) | 18 (8.9%) |
| **Transplant Era** | | | | |
| 1/1/1987–12/31/1999 | 18589 (42.0%) | 2364 (24.6%) | 249 (26.6%) | 30 (14.8%) |
| 1/1/2000–12/31/2009 | 12734 (28.8%) | 2861 (29.8%) | 310 (33.1%) | 59 (29.1%) |

*(Continued)*

**Table 1.** (Continued)

| | | | | |
|---|---|---|---|---|
| 1/1/2010–12/31/2018 | 12941 (29.2%) | 4367 (45.5%) | 377 (40.3%) | 114 (56.2%) |
| **Hospitalization Status** | | | | |
| ICU | 16392 (37.0%) | 3493 (36.4%) | 374 (40.0%) | 74 (36.5%) |
| Hospitalized non-ICU | 6297 (14.2%) | 1573 (16.4%) | 160 (17.1%) | 34 (16.7%) |
| Not hospitalized | 21489 (48.5%) | 4517 (47.1%) | 402 (42.9%) | 95 (46.8%) |
| **Functional Status** | | | | |
| Moribund/hospitalized/severely disabled | 0 (0%) | 0 (0%) | 0 (0%) | 0 (0%) |
| Requires significant assistance | 0 (0%) | 0 (0%) | 0 (0%) | 0 (0%) |
| Normal with minor/no symptoms | 0 (0%) | 0 (0%) | 0 (0%) | 0 (0%) |
| **Received VAD** | | | | |
| No | 23208 (52.4%) | 5582 (58.2%) | 505 (54.0%) | 102 (50.2%) |
| Yes | 6190 (14.0%) | 1923 (20.0%) | 260 (27.8%) | 69 (34.0%) |
| **Received Life Support** | | | | |
| No | 17136 (38.7%) | 2456 (25.6%) | 182 (19.4%) | 19 (9.4%) |
| Yes | 27043 (61.1%) | 7125 (74.3%) | 754 (80.6%) | 184 (90.6%) |
| **Received ECMO** | | | | |
| No | 44096 (99.6%) | 9553 (99.6%) | 922 (98.5%) | 199 (98.0%) |
| Yes | 168 (0.4%) | 39 (0.4%) | 14 (1.5%) | 4 (2.0%) |
| **Total Ischemic Time (minutes)** | | | | |
| Mean (SD) | 180 (62.2) | 181 (61.1) | 196 (65.8) | 190 (69.3) |
| Median [Min, Max] | 179 [14.0, 720] | 180 [13.0, 720] | 194 [23.0, 488] | 189 [56.0, 524] |
| **PA Systolic Blood Pressure (mm/Hg)** | | | | |
| Mean (SD) | 41.7 (14.5) | 44.0 (14.2) | 42.1 (14.2) | 42.4 (14.6) |
| Median [Min, Max] | 40.0 [1.00, 175] | 43.0 [5.00, 132] | 41.0 [5.00, 85.0] | 40.0 [9.00, 80.0] |
| **Number of HLA-AB Mismatches** | | | | |
| 0 | 59 (0.1%) | 12 (0.1%) | 2 (0.2%) | 0 (0%) |
| 1 | 5764 (13.0%) | 799 (8.3%) | 136 (14.5%) | 13 (6.4%) |
| 2 | 27631 (62.4%) | 7242 (75.5%) | 653 (69.8%) | 164 (80.8%) |
| **Number of HL-DR Mismatches** | | | | |
| 0 | 2013 (4.5%) | 368 (3.8%) | 52 (5.6%) | 6 (3.0%) |
| 1 | 14310 (32.3%) | 3288 (34.3%) | 349 (37.3%) | 72 (35.5%) |
| 2 | 17234 (38.9%) | 4421 (46.1%) | 394 (42.1%) | 99 (48.8%) |
| **PRA > 10%** | | | | |
| No | 35187 (79.5%) | 7004 (73.0%) | 694 (74.1%) | 150 (73.9%) |
| Yes | 5088 (11.5%) | 1742 (18.2%) | 147 (15.7%) | 40 (19.7%) |
| **Creatinine (mg/dL)** | | | | |
| Mean (SD) | 1.34 (1.02) | 1.49 (1.17) | 1.39 (0.925) | 1.65 (1.31) |
| Median [Min, Max] | 1.20 [0.100, 50.0] | 1.30 [0.100, 25.1] | 1.20 [0.200, 14.0] | 1.30 [0.300, 12.0] |
| **Total Bilirubin (mg/dL)** | | | | |
| Mean (SD) | 1.16 (2.51) | 1.14 (2.27) | 1.31 (2.24) | 1.28 (1.43) |
| Median [Min, Max] | 0.800 [0.100, 86.0] | 0.700 [0.100, 69.0] | 0.800 [0.100, 35.7] | 0.900 [0.200, 10.3] |
| **Received Inotropes** | | | | |
| No | 27886 (63.0%) | 5584 (58.2%) | 563 (60.1%) | 130 (64.0%) |
| Yes | 16378 (37.0%) | 4008 (41.8%) | 373 (39.9%) | 73 (36.0%) |
| **Type of Follow-up Care** | | | | |
| Transplant center | 15011 (33.9%) | 4177 (43.5%) | 308 (32.9%) | 81 (39.9%) |
| Non transplant center specialist | 436 (1.0%) | 99 (1.0%) | 15 (1.6%) | 1 (0.5%) |
| PCP | 203 (0.5%) | 18 (0.2%) | 5 (0.5%) | 0 (0%) |

(*Continued*)

**Table 1.** (Continued)

| | No Perioperative Stroke | | Perioperative Stroke | | Overall | |
|---|---|---|---|---|---|---|
| Other | 4308 (9.7%) | 665 (6.9%) | 31 (3.3%) | 5 (2.5%) | | |
| **Experienced Follow-up Hospitalizations** | | | | | | |
| No | 7273 (16.4%) | 2018 (21.0%) | 138 (14.7%) | 33 (16.3%) | | |
| Yes | 9599 (21.7%) | 2480 (25.9%) | 190 (20.3%) | 49 (24.1%) | | |
| **Wait Time (Days)** | | | | | | |
| Mean (SD) | 211 (329) | 211 (337) | 228 (354) | 211 (275) | | |
| Median [Min, Max] | 96.0 [0, 6410] | 87.0 [0, 5150] | 101 [0, 3880] | 92.0 [0, 1410] | | |

| | No Perioperative Stroke | | Perioperative Stroke | | Overall | |
|---|---|---|---|---|---|---|
| | White (N = 44264) | Black (N = 9592) | White (N = 936) | Black (N = 203) | White (N = 45200) | Black (N = 9795) |
| **Recipient Age** | | | | | | |
| Mean (SD) | 53.8 (11.4) | 49.0 (12.6) | 55.4 (10.7) | 50.7 (12.9) | 53.8 (11.4) | 49.0 (12.7) |
| Median [Min, Max] | 56.1 [18.0, 78.6] | 51.0 [18.0, 76.4] | 57.4 [18.5, 79.0] | 52.1 [18.7, 78.4] | 56.2 [18.0, 79.0] | 51.1 [18.0, 78.4] |
| **Recipient Sex** | | | | | | |
| Male | 34947 (79.0%) | 6354 (66.2%) | 701 (74.9%) | 132 (65.0%) | 35648 (78.9%) | 6486 (66.2%) |
| Female | 9317 (21.0%) | 3238 (33.8%) | 235 (25.1%) | 71 (35.0%) | 9552 (21.1%) | 3309 (33.8%) |
| **Education Level** | | | | | | |
| Below high school | 794 (1.8%) | 234 (2.4%) | 27 (2.9%) | 8 (3.9%) | 821 (1.8%) | 242 (2.5%) |
| High school | 11727 (26.5%) | 3613 (37.7%) | 325 (34.7%) | 79 (38.9%) | 12052 (26.7%) | 3692 (37.7%) |
| College and above | 15601 (35.2%) | 3504 (36.5%) | 430 (45.9%) | 89 (43.8%) | 16031 (35.5%) | 3593 (36.7%) |
| Missing | 16142 (36.5%) | 2241 (23.4%) | 154 (16.5%) | 27 (13.3%) | 16296 (36.1%) | 2268 (23.2%) |
| **Insurance** | | | | | | |
| Medicare | 2726 (6.2%) | 1659 (17.3%) | 64 (6.8%) | 38 (18.7%) | 2790 (6.2%) | 1697 (17.3%) |
| Medicaid | 9302 (21.0%) | 2513 (26.2%) | 258 (27.6%) | 73 (36.0%) | 9560 (21.2%) | 2586 (26.4%) |
| Private | 21110 (47.7%) | 3849 (40.1%) | 555 (59.3%) | 87 (42.9%) | 21665 (47.9%) | 3936 (40.2%) |
| Other | 1560 (3.5%) | 494 (5.2%) | 41 (4.4%) | 5 (2.5%) | 1601 (3.5%) | 499 (5.1%) |
| Missing | 9566 (21.6%) | 1077 (11.2%) | 18 (1.9%) | 0 (0%) | 9584 (21.2%) | 1077 (11.0%) |
| **BMI Categories** | | | | | | |
| Underweight | 797 (1.8%) | 228 (2.4%) | 12 (1.3%) | 6 (3.0%) | 809 (1.8%) | 234 (2.4%) |
| Normal or health weight | 16797 (37.9%) | 3449 (36.0%) | 348 (37.2%) | 70 (34.5%) | 17145 (37.9%) | 3519 (35.9%) |
| Overweight | 16337 (36.9%) | 3227 (33.6%) | 346 (37.0%) | 71 (35.0%) | 16683 (36.9%) | 3298 (33.7%) |
| Obese | 9492 (21.4%) | 2565 (26.7%) | 218 (23.3%) | 55 (27.1%) | 9710 (21.5%) | 2620 (26.7%) |
| Missing | 841 (1.9%) | 123 (1.3%) | 12 (1.3%) | 1 (0.5%) | 853 (1.9%) | 124 (1.3%) |
| **Malignancy** | | | | | | |
| No | 31119 (70.3%) | 7886 (82.2%) | 825 (88.1%) | 192 (94.6%) | 31944 (70.7%) | 8078 (82.5%) |
| Yes | 2166 (4.9%) | 456 (4.8%) | 70 (7.5%) | 9 (4.4%) | 2236 (4.9%) | 465 (4.7%) |
| Missing | 10979 (24.8%) | 1250 (13.0%) | 41 (4.4%) | 2 (1.0%) | 11020 (24.4%) | 1252 (12.8%) |
| **Smoker** | | | | | | |
| No | 12299 (27.8%) | 4156 (43.3%) | 310 (33.1%) | 109 (53.7%) | 12609 (27.9%) | 4265 (43.5%) |
| Yes | 16624 (37.6%) | 3409 (35.5%) | 471 (50.3%) | 75 (36.9%) | 17095 (37.8%) | 3484 (35.6%) |
| Missing | 15341 (34.7%) | 2027 (21.1%) | 155 (16.6%) | 19 (9.4%) | 15496 (34.3%) | 2046 (20.9%) |
| **Hypertension** | | | | | | |
| No | 15424 (34.8%) | 2954 (30.8%) | 370 (39.5%) | 61 (30.0%) | 15794 (34.9%) | 3015 (30.8%) |
| Yes | 11239 (25.4%) | 3155 (32.9%) | 322 (34.4%) | 82 (40.4%) | 11561 (25.6%) | 3237 (33.0%) |
| Missing | 17601 (39.8%) | 3483 (36.3%) | 244 (26.1%) | 60 (29.6%) | 17845 (39.5%) | 3543 (36.2%) |
| **Diabetes Mellitus** | | | | | | |
| No | 36570 (82.6%) | 7497 (78.2%) | 696 (74.4%) | 139 (68.5%) | 37266 (82.4%) | 7636 (78.0%) |
| Yes | 7694 (17.4%) | 2095 (21.8%) | 240 (25.6%) | 64 (31.5%) | 7934 (17.6%) | 2159 (22.0%) |

*(Continued)*

**Table 1.** (Continued)

| | | | | | | |
|---|---|---|---|---|---|---|
| **Prior Stroke** | | | | | | |
| No | 42189 (95.3%) | 8972 (93.5%) | 800 (85.5%) | 170 (83.7%) | 42989 (95.1%) | 9142 (93.3%) |
| Yes | 2075 (4.7%) | 620 (6.5%) | 136 (14.5%) | 33 (16.3%) | 2211 (4.9%) | 653 (6.7%) |
| **History of Cerebrovascular Disease** | | | | | | |
| No | 31868 (72.0%) | 7856 (81.9%) | 835 (89.2%) | 177 (87.2%) | 32703 (72.4%) | 8033 (82.0%) |
| Yes | 1496 (3.4%) | 488 (5.1%) | 62 (6.6%) | 21 (10.3%) | 1558 (3.4%) | 509 (5.2%) |
| Missing | 10900 (24.6%) | 1248 (13.0%) | 39 (4.2%) | 5 (2.5%) | 10939 (24.2%) | 1253 (12.8%) |
| **Cardiovascular Etiology** | | | | | | |
| Ischemic | 22291 (50.4%) | 2112 (22.0%) | 517 (55.2%) | 47 (23.2%) | 22808 (50.5%) | 2159 (22.0%) |
| Non-ischemic | 17527 (39.6%) | 6776 (70.6%) | 317 (33.9%) | 138 (68.0%) | 17844 (39.5%) | 6914 (70.6%) |
| Other | 4442 (10.0%) | 704 (7.3%) | 102 (10.9%) | 18 (8.9%) | 4544 (10.1%) | 722 (7.4%) |
| Missing | 4 (0.0%) | 0 (0%) | 0 (0%) | 0 (0%) | 4 (0.0%) | 0 (0%) |
| **Transplant Era** | | | | | | |
| 1/1/2000–12/31/2005 | 18589 (42.0%) | 2364 (24.6%) | 249 (26.6%) | 30 (14.8%) | 18838 (41.7%) | 2394 (24.4%) |
| 1/1/2006–12/31/2011 | 12734 (28.8%) | 2861 (29.8%) | 310 (33.1%) | 59 (29.1%) | 13044 (28.9%) | 2920 (29.8%) |
| 1/1/2012–12/31/2018 | 12941 (29.2%) | 4367 (45.5%) | 377 (40.3%) | 114 (56.2%) | 13318 (29.5%) | 4481 (45.7%) |
| **Hospitalization Status** | | | | | | |
| ICU | 16392 (37.0%) | 3493 (36.4%) | 374 (40.0%) | 74 (36.5%) | 16766 (37.1%) | 3567 (36.4%) |
| Hospitalized non-ICU | 6297 (14.2%) | 1573 (16.4%) | 160 (17.1%) | 34 (16.7%) | 6457 (14.3%) | 1607 (16.4%) |
| Not hospitalized | 21489 (48.5%) | 4517 (47.1%) | 402 (42.9%) | 95 (46.8%) | 21891 (48.4%) | 4612 (47.1%) |
| Missing | 86 (0.2%) | 9 (0.1%) | 0 (0%) | 0 (0%) | 86 (0.2%) | 9 (0.1%) |
| **Functional Status** | | | | | | |
| Moribund/hospitalized/severely disabled | 0 (0%) | 0 (0%) | 0 (0%) | 0 (0%) | 0 (0%) | 0 (0%) |
| Requires significant assistance | 0 (0%) | 0 (0%) | 0 (0%) | 0 (0%) | 0 (0%) | 0 (0%) |
| Normal with minor/no symptoms | 0 (0%) | 0 (0%) | 0 (0%) | 0 (0%) | 0 (0%) | 0 (0%) |
| Missing | 44264 (100%) | 9592 (100%) | 936 (100%) | 203 (100%) | 45200 (100%) | 9795 (100%) |
| **Received VAD** | | | | | | |
| No | 23208 (52.4%) | 5582 (58.2%) | 505 (54.0%) | 102 (50.2%) | 23713 (52.5%) | 5684 (58.0%) |
| Yes | 6190 (14.0%) | 1923 (20.0%) | 260 (27.8%) | 69 (34.0%) | 6450 (14.3%) | 1992 (20.3%) |
| Missing | 14866 (33.6%) | 2087 (21.8%) | 171 (18.3%) | 32 (15.8%) | 15037 (33.3%) | 2119 (21.6%) |
| **Received Life Support** | | | | | | |
| No | 17136 (38.7%) | 2456 (25.6%) | 182 (19.4%) | 19 (9.4%) | 17318 (38.3%) | 2475 (25.3%) |
| Yes | 27043 (61.1%) | 7125 (74.3%) | 754 (80.6%) | 184 (90.6%) | 27797 (61.5%) | 7309 (74.6%) |
| Missing | 85 (0.2%) | 11 (0.1%) | 0 (0%) | 0 (0%) | 85 (0.2%) | 11 (0.1%) |
| **Received ECMO** | | | | | | |
| No | 44096 (99.6%) | 9553 (99.6%) | 922 (98.5%) | 199 (98.0%) | 45018 (99.6%) | 9752 (99.6%) |
| Yes | 168 (0.4%) | 39 (0.4%) | 14 (1.5%) | 4 (2.0%) | 182 (0.4%) | 43 (0.4%) |
| **Total Ischemic Time (minutes)** | | | | | | |
| Mean (SD) | 180 (62.2) | 181 (61.1) | 196 (65.8) | 190 (69.3) | 180 (62.3) | 181 (61.3) |
| Median [Min, Max] | 179 [14.0, 720] | 180 [13.0, 720] | 194 [23.0, 488] | 189 [56.0, 524] | 179 [14.0, 720] | 180 [13.0, 720] |
| Missing | 1707 (3.9%) | 343 (3.6%) | 40 (4.3%) | 7 (3.4%) | 1747 (3.9%) | 350 (3.6%) |
| **PA Systolic Blood Pressure (mm/Hg)** | | | | | | |
| Mean (SD) | 41.7 (14.5) | 44.0 (14.2) | 42.1 (14.2) | 42.4 (14.6) | 41.7 (14.5) | 43.9 (14.2) |
| Median [Min, Max] | 40.0 [1.00, 175] | 43.0 [5.00, 132] | 41.0 [5.00, 85.0] | 40.0 [9.00, 80.0] | 40.0 [1.00, 175] | 43.0 [5.00, 132] |
| Missing | 11924 (26.9%) | 1429 (14.9%) | 85 (9.1%) | 14 (6.9%) | 12009 (26.6%) | 1443 (14.7%) |
| **Number of HLA-AB Mismatches** | | | | | | |
| 0 | 59 (0.1%) | 12 (0.1%) | 2 (0.2%) | 0 (0%) | 61 (0.1%) | 12 (0.1%) |
| 1 | 5764 (13.0%) | 799 (8.3%) | 136 (14.5%) | 13 (6.4%) | 5900 (13.1%) | 812 (8.3%) |

*(Continued)*

**Table 1.** (Continued)

| | | | | | | |
|---|---|---|---|---|---|---|
| 2 | 27631 (62.4%) | 7242 (75.5%) | 653 (69.8%) | 164 (80.8%) | 28284 (62.6%) | 7406 (75.6%) |
| Missing | 10810 (24.4%) | 1539 (16.0%) | 145 (15.5%) | 26 (12.8%) | 10955 (24.2%) | 1565 (16.0%) |
| **Number of HL-DR Mismatches** | | | | | | |
| 0 | 2013 (4.5%) | 368 (3.8%) | 52 (5.6%) | 6 (3.0%) | 2065 (4.6%) | 374 (3.8%) |
| 1 | 14310 (32.3%) | 3288 (34.3%) | 349 (37.3%) | 72 (35.5%) | 14659 (32.4%) | 3360 (34.3%) |
| 2 | 17234 (38.9%) | 4421 (46.1%) | 394 (42.1%) | 99 (48.8%) | 17628 (39.0%) | 4520 (46.1%) |
| Missing | 10707 (24.2%) | 1515 (15.8%) | 141 (15.1%) | 26 (12.8%) | 10848 (24.0%) | 1541 (15.7%) |
| **PRA > 10%** | | | | | | |
| No | 35187 (79.5%) | 7004 (73.0%) | 694 (74.1%) | 150 (73.9%) | 35881 (79.4%) | 7154 (73.0%) |
| Yes | 5088 (11.5%) | 1742 (18.2%) | 147 (15.7%) | 40 (19.7%) | 5235 (11.6%) | 1782 (18.2%) |
| Missing | 3989 (9.0%) | 846 (8.8%) | 95 (10.1%) | 13 (6.4%) | 4084 (9.0%) | 859 (8.8%) |
| **Creatinine (mg/dL)** | | | | | | |
| Mean (SD) | 1.34 (1.02) | 1.49 (1.17) | 1.39 (0.925) | 1.65 (1.31) | 1.34 (1.02) | 1.49 (1.17) |
| Median [Min, Max] | 1.20 [0.100, 50.0] | 1.30 [0.100, 25.1] | 1.20 [0.200, 14.0] | 1.30 [0.300, 12.0] | 1.20 [0.100, 50.0] | 1.30 [0.100, 25.1] |
| Missing | 9132 (20.6%) | 1024 (10.7%) | 20 (2.1%) | 2 (1.0%) | 9152 (20.2%) | 1026 (10.5%) |
| **Total Bilirubin (mg/dL)** | | | | | | |
| Mean (SD) | 1.16 (2.51) | 1.14 (2.27) | 1.31 (2.24) | 1.28 (1.43) | 1.17 (2.50) | 1.14 (2.25) |
| Median [Min, Max] | 0.800 [0.100, 86.0] | 0.700 [0.100, 69.0] | 0.800 [0.100, 35.7] | 0.900 [0.200, 10.3] | 0.800 [0.100, 86.0] | 0.700 [0.100, 69.0] |
| Missing | 10869 (24.6%) | 1286 (13.4%) | 33 (3.5%) | 6 (3.0%) | 10902 (24.1%) | 1292 (13.2%) |
| **Received Inotropes** | | | | | | |
| No | 27886 (63.0%) | 5584 (58.2%) | 563 (60.1%) | 130 (64.0%) | 28449 (62.9%) | 5714 (58.3%) |
| Yes | 16378 (37.0%) | 4008 (41.8%) | 373 (39.9%) | 73 (36.0%) | 16751 (37.1%) | 4081 (41.7%) |
| **Type of Follow-up Care** | | | | | | |
| Transplant center | 15011 (33.9%) | 4177 (43.5%) | 308 (32.9%) | 81 (39.9%) | 15319 (33.9%) | 4258 (43.5%) |
| Non transplant center specialist | 436 (1.0%) | 99 (1.0%) | 15 (1.6%) | 1 (0.5%) | 451 (1.0%) | 100 (1.0%) |
| PCP | 203 (0.5%) | 18 (0.2%) | 5 (0.5%) | 0 (0%) | 208 (0.5%) | 18 (0.2%) |
| Other | 4308 (9.7%) | 665 (6.9%) | 31 (3.3%) | 5 (2.5%) | 4339 (9.6%) | 670 (6.8%) |
| Missing | 24306 (54.9%) | 4633 (48.3%) | 577 (61.6%) | 116 (57.1%) | 24883 (55.1%) | 4749 (48.5%) |
| **Experienced Follow-up Hospitalizations** | | | | | | |
| No | 7273 (16.4%) | 2018 (21.0%) | 138 (14.7%) | 33 (16.3%) | 7411 (16.4%) | 2051 (20.9%) |
| Yes | 9599 (21.7%) | 2480 (25.9%) | 190 (20.3%) | 49 (24.1%) | 9789 (21.7%) | 2529 (25.8%) |
| Missing | 27392 (61.9%) | 5094 (53.1%) | 608 (65.0%) | 121 (59.6%) | 28000 (61.9%) | 5215 (53.2%) |
| **Wait Time (Days)** | | | | | | |
| Mean (SD) | 211 (329) | 211 (337) | 228 (354) | 211 (275) | 212 (330) | 211 (336) |
| Median [Min, Max] | 96.0 [0, 6410] | 87.0 [0, 5150] | 101 [0, 3880] | 92.0 [0, 1410] | 96.0 [0, 6410] | 87.0 [0, 5150] |
| Missing | 1 (0.0%) | 1 (0.0%) | 0 (0%) | 0 (0%) | 1 (0.0%) | 1 (0.0% |

% of missing data can be calculated from the proportions displayed in this table

ECMO: extracorporeal membrane oxygenation

VAD: ventricular assist device

$p = 0.04$) (Fig 1). In age- and sex-adjusted models, there was no evidence of racial disparities in mortality within 6 months of post-transplant stroke (HR = 0.99, 95% CI: 0.76–1.28) but beyond 6 months Black post-transplant stroke survivors had a 44% (HR = 1.44, 95% CI: 1.07–1.94) higher mortality rate than white post-transplant stroke survivors. Further adjusting for covariates strengthened this association (HR = 1.57, 95% CI: 1.12–2.18; Table 2, Model 1), but additionally adjusting for education and insurance did not change this association (HR = 1.57, 95% CI: 1.11–2.21; Table 3, Model 2). Additionally including the setting of long-term post-

**Table 2. Heart transplant recipient characteristics by race and transplant era for post-transplant stroke survivors, Scientific Registry of Transplant Recipients (SRTR), January 1, 1987 to December 31, 2018; n (%) unless otherwise specified.**

| | White | | | Black | | |
|---|---|---|---|---|---|---|
| | 1/1/1987–12/31/1999 (N = 249) | 1/1/2000–12/31/2009 (N = 310) | 1/1/2010–12/31/2018 (N = 377) | 1/1/1987–12/31/1999 (N = 30) | 1/1/2000–12/31/2009 (N = 59) | 1/1/2010–12/31/2018 (N = 114) |
| **Recipient Age** | | | | | | |
| Mean (SD) | 54.4 (9.41) | 55.0 (10.7) | 56.4 (11.5) | 48.1 (12.2) | 47.5 (13.7) | 53.1 (12.3) |
| Median [Min, Max] | 56.1 [22.8, 73.3] | 57.2 [19.3, 73.8] | 59.1 [18.5, 79.0] | 49.5 [22.8, 69.4] | 49.3 [20.7, 69.3] | 53.8 [18.7, 78.4] |
| **Recipient Sex** | | | | | | |
| Male | 185 (74.3%) | 235 (75.8%) | 281 (74.5%) | 20 (66.7%) | 30 (50.8%) | 82 (71.9%) |
| Female | 64 (25.7%) | 75 (24.2%) | 96 (25.5%) | 10 (33.3%) | 29 (49.2%) | 32 (28.1%) |
| **Education Level** | | | | | | |
| Below high school | 10 (4.0%) | 10 (3.2%) | 7 (1.9%) | 2 (6.7%) | 1 (1.7%) | 5 (4.4%) |
| High school | 90 (36.1%) | 102 (32.9%) | 133 (35.3%) | 12 (40.0%) | 16 (27.1%) | 51 (44.7%) |
| College and above | 69 (27.7%) | 135 (43.5%) | 226 (59.9%) | 8 (26.7%) | 27 (45.8%) | 54 (47.4%) |
| **Insurance** | | | | | | |
| Medicare | 17 (6.8%) | 21 (6.8%) | 26 (6.9%) | 7 (23.3%) | 12 (20.3%) | 19 (16.7%) |
| Medicaid | 53 (21.3%) | 74 (23.9%) | 131 (34.7%) | 11 (36.7%) | 21 (35.6%) | 41 (36.0%) |
| Private | 150 (60.2%) | 194 (62.6%) | 211 (56.0%) | 11 (36.7%) | 24 (40.7%) | 52 (45.6%) |
| Other | 11 (4.4%) | 21 (6.8%) | 9 (2.4%) | 1 (3.3%) | 2 (3.4%) | 2 (1.8%) |
| **BMI Categories** | | | | | | |
| Underweight | 3 (1.2%) | 6 (1.9%) | 3 (0.8%) | 3 (10.0%) | 2 (3.4%) | 1 (0.9%) |
| Normal or health weight | 101 (40.6%) | 122 (39.4%) | 125 (33.2%) | 12 (40.0%) | 21 (35.6%) | 37 (32.5%) |
| Overweight | 92 (36.9%) | 117 (37.7%) | 137 (36.3%) | 10 (33.3%) | 21 (35.6%) | 40 (35.1%) |
| Obese | 45 (18.1%) | 64 (20.6%) | 109 (28.9%) | 4 (13.3%) | 15 (25.4%) | 36 (31.6%) |
| **Malignancy** | | | | | | |
| No | 204 (81.9%) | 284 (91.6%) | 337 (89.4%) | 27 (90.0%) | 57 (96.6%) | 108 (94.7%) |
| Yes | 11 (4.4%) | 19 (6.1%) | 40 (10.6%) | 1 (3.3%) | 2 (3.4%) | 6 (5.3%) |
| **Smoker** | | | | | | |
| No | 6 (2.4%) | 124 (40.0%) | 180 (47.7%) | 2 (6.7%) | 36 (61.0%) | 71 (62.3%) |
| Yes | 116 (46.6%) | 158 (51.0%) | 197 (52.3%) | 11 (36.7%) | 21 (35.6%) | 43 (37.7%) |
| **Hypertension** | | | | | | |
| No | 126 (50.6%) | 179 (57.7%) | 65 (17.2%) | 19 (63.3%) | 28 (47.5%) | 14 (12.3%) |
| Yes | 90 (36.1%) | 120 (38.7%) | 112 (29.7%) | 9 (30.0%) | 27 (45.8%) | 46 (40.4%) |
| **Diabetes Mellitus** | | | | | | |
| No | 200 (80.3%) | 238 (76.8%) | 258 (68.4%) | 23 (76.7%) | 38 (64.4%) | 78 (68.4%) |
| Yes | 49 (19.7%) | 72 (23.2%) | 119 (31.6%) | 7 (23.3%) | 21 (35.6%) | 36 (31.6%) |
| **Prior Stroke** | | | | | | |
| No | 210 (84.3%) | 260 (83.9%) | 330 (87.5%) | 25 (83.3%) | 48 (81.4%) | 97 (85.1%) |
| Yes | 39 (15.7%) | 50 (16.1%) | 47 (12.5%) | 5 (16.7%) | 11 (18.6%) | 17 (14.9%) |
| **History of Cerebrovascular Disease** | | | | | | |
| No | 203 (81.5%) | 286 (92.3%) | 346 (91.8%) | 26 (86.7%) | 54 (91.5%) | 97 (85.1%) |
| Yes | 15 (6.0%) | 18 (5.8%) | 29 (7.7%) | 2 (6.7%) | 5 (8.5%) | 14 (12.3%) |
| **Cardiovascular Etiology** | | | | | | |
| Ischemic | 158 (63.5%) | 181 (58.4%) | 178 (47.2%) | 10 (33.3%) | 10 (16.9%) | 27 (23.7%) |
| Non-ischemic | 76 (30.5%) | 93 (30.0%) | 148 (39.3%) | 19 (63.3%) | 45 (76.3%) | 74 (64.9%) |
| Other | 15 (6.0%) | 36 (11.6%) | 51 (13.5%) | 1 (3.3%) | 4 (6.8%) | 13 (11.4%) |
| **Hospitalization Status** | | | | | | |
| ICU | 141 (56.6%) | 116 (37.4%) | 117 (31.0%) | 20 (66.7%) | 27 (45.8%) | 27 (23.7%) |

*(Continued)*

**Table 2.** (Continued)

| | White | | | Black | | |
|---|---|---|---|---|---|---|
| | 1/1/1987–12/31/1999 (N = 249) | 1/1/2000–12/31/2009 (N = 310) | 1/1/2010–12/31/2018 (N = 377) | 1/1/1987–12/31/1999 (N = 30) | 1/1/2000–12/31/2009 (N = 59) | 1/1/2010–12/31/2018 (N = 114) |
| Hospitalized non-ICU | 34 (13.7%) | 60 (19.4%) | 66 (17.5%) | 4 (13.3%) | 12 (20.3%) | 18 (15.8%) |
| Not hospitalized | 74 (29.7%) | 134 (43.2%) | 194 (51.5%) | 6 (20.0%) | 20 (33.9%) | 69 (60.5%) |
| **Received VAD** | | | | | | |
| No | 100 (40.2%) | 182 (58.7%) | 223 (59.2%) | 9 (30.0%) | 39 (66.1%) | 54 (47.4%) |
| Yes | 27 (10.8%) | 79 (25.5%) | 154 (40.8%) | 2 (6.7%) | 7 (11.9%) | 60 (52.6%) |
| **Received Life Support** | | | | | | |
| No | 67 (26.9%) | 76 (24.5%) | 39 (10.3%) | 6 (20.0%) | 7 (11.9%) | 6 (5.3%) |
| Yes | 182 (73.1%) | 234 (75.5%) | 338 (89.7%) | 24 (80.0%) | 52 (88.1%) | 108 (94.7%) |
| **Received ECMO** | | | | | | |
| No | 246 (98.8%) | 305 (98.4%) | 371 (98.4%) | 30 (100%) | 56 (94.9%) | 113 (99.1%) |
| Yes | 3 (1.2%) | 5 (1.6%) | 6 (1.6%) | 0 (0%) | 3 (5.1%) | 1 (0.9%) |
| **Total Ischemic Time (minutes)** | | | | | | |
| Mean (SD) | 185 (62.0) | 199 (64.4) | 202 (68.5) | 162 (57.1) | 226 (64.1) | 180 (68.2) |
| Median [Min, Max] | 184 [42.0, 458] | 197 [69.0, 390] | 200 [23.0, 488] | 155 [56.0, 248] | 219 [91.0, 383] | 174 [73.0, 524] |
| **PA Systolic Blood Pressure (mm/Hg)** | | | | | | |
| Mean (SD) | 46.5 (14.9) | 41.7 (13.7) | 39.6 (13.5) | 54.6 (13.5) | 44.8 (13.8) | 38.4 (13.4) |
| Median [Min, Max] | 46.0 [15.0, 84.0] | 40.0 [13.0, 85.0] | 38.0 [5.00, 78.0] | 52.0 [30.0, 80.0] | 45.0 [15.0, 72.0] | 35.0 [9.00, 72.0] |
| **Number of HLA-AB Mismatches** | | | | | | |
| 0 | 1 (0.4%) | 0 (0%) | 1 (0.3%) | 0 (0%) | 0 (0%) | 0 (0%) |
| 1 | 42 (16.9%) | 38 (12.3%) | 56 (14.9%) | 4 (13.3%) | 3 (5.1%) | 6 (5.3%) |
| 2 | 157 (63.1%) | 218 (70.3%) | 278 (73.7%) | 22 (73.3%) | 45 (76.3%) | 97 (85.1%) |
| **Number of HL-DR Mismatches** | | | | | | |
| 0 | 16 (6.4%) | 19 (6.1%) | 17 (4.5%) | 2 (6.7%) | 1 (1.7%) | 3 (2.6%) |
| 1 | 95 (38.2%) | 106 (34.2%) | 148 (39.3%) | 10 (33.3%) | 19 (32.2%) | 43 (37.7%) |
| 2 | 92 (36.9%) | 132 (42.6%) | 170 (45.1%) | 14 (46.7%) | 28 (47.5%) | 57 (50.0%) |
| **PRA > 10%** | | | | | | |
| No | 206 (82.7%) | 250 (80.6%) | 238 (63.1%) | 27 (90.0%) | 45 (76.3%) | 78 (68.4%) |
| Yes | 29 (11.6%) | 35 (11.3%) | 83 (22.0%) | 3 (10.0%) | 10 (16.9%) | 27 (23.7%) |
| **Creatinine (mg/dL)** | | | | | | |
| Mean (SD) | 1.45 (0.865) | 1.47 (1.15) | 1.29 (0.733) | 1.79 (2.09) | 1.37 (0.514) | 1.75 (1.34) |
| Median [Min, Max] | 1.20 [0.400, 7.70] | 1.27 [0.300, 14.0] | 1.17 [0.200, 8.50] | 1.30 [0.400, 12.0] | 1.20 [0.600, 3.10] | 1.43 [0.300, 11.4] |
| **Total Bilirubin (mg/dL)** | | | | | | |
| Mean (SD) | 1.25 (1.73) | 1.43 (2.23) | 1.24 (2.51) | 1.70 (2.37) | 1.56 (1.50) | 1.03 (0.976) |
| Median [Min, Max] | 0.800 [0.200, 17.6] | 0.900 [0.100, 28.0] | 0.700 [0.100, 35.7] | 0.900 [0.300, 10.3] | 1.10 [0.200, 9.70] | 0.800 [0.200, 7.90] |
| **Received Inotropes** | | | | | | |
| No | 116 (46.6%) | 175 (56.5%) | 272 (72.1%) | 11 (36.7%) | 35 (59.3%) | 84 (73.7%) |
| Yes | 133 (53.4%) | 135 (43.5%) | 105 (27.9%) | 19 (63.3%) | 24 (40.7%) | 30 (26.3%) |
| **Type of Follow-up Care** | | | | | | |
| Transplant center | 52 (20.9%) | 56 (18.1%) | 200 (53.1%) | 13 (43.3%) | 13 (22.0%) | 55 (48.2%) |
| Non transplant center specialist | 4 (1.6%) | 4 (1.3%) | 7 (1.9%) | 0 (0%) | 0 (0%) | 1 (0.9%) |
| PCP | 3 (1.2%) | 2 (0.6%) | 0 (0%) | 0 (0%) | 0 (0%) | 0 (0%) |

*(Continued)*

**Table 2.** (Continued)

| | White | | | Black | | |
|---|---|---|---|---|---|---|
| | 1/1/1987–12/31/1999 (N = 249) | 1/1/2000–12/31/2009 (N = 310) | 1/1/2010–12/31/2018 (N = 377) | 1/1/1987–12/31/1999 (N = 30) | 1/1/2000–12/31/2009 (N = 59) | 1/1/2010–12/31/2018 (N = 114) |
| Other | 31 (12.4%) | 0 (0%) | 0 (0%) | 5 (16.7%) | 0 (0%) | 0 (0%) |
| **Experienced Follow-up Hospitalizations** | | | | | | |
| No | 25 (10.0%) | 11 (3.5%) | 102 (27.1%) | 4 (13.3%) | 1 (1.7%) | 28 (24.6%) |
| Yes | 47 (18.9%) | 41 (13.2%) | 102 (27.1%) | 12 (40.0%) | 10 (16.9%) | 27 (23.7%) |
| **Wait Time (Days)** | | | | | | |
| Mean (SD) | 175 (213) | 225 (372) | 264 (405) | 118 (114) | 187 (270) | 247 (301) |
| Median [Min, Max] | 96.0 [0, 1260] | 87.0 [1.00, 2740] | 118 [1.00, 3880] | 77.0 [10.0, 516] | 82.0 [3.00, 1410] | 109 [0, 1230] |

% of missing data can be calculated from the proportions displayed in this table

ECMO: extracorporeal membrane oxygenation

VAD: ventricular assist device

transplant care as a covariate somewhat attenuated the association to 1.45 (95% CI: 1.03–2.04). In a sensitivity analysis, stratifying by transplant center did not materially alter these results.

The survival probability difference between white and black post-transplant stroke survivors appear to narrow in successive transplantation eras (Fig 2). The higher mortality among

**Table 3.** Hazard ratios and 95% confidence intervals comparing mortality rates for black post-transplant stroke survivors to mortality rates for white post-transplant stroke survivors according to era, Scientific Registry of Transplant Recipients (SRTR), January 1, 1987 To December 31, 2018.

| Time Frame | Cases/Person-years | | Model 1* | Model 2** | Model 3† |
|---|---|---|---|---|---|
| | Black | White | | | |
| **Overall** | 127/655 | 599/4,169 | 1.22 (1.00, 1.52) | 1.21 (0.97, 1.50) | 1.19 (0.96, 1.48) |
| **Within 6 Months** | | | | | |
| All eras | 73/71 | 335/330 | 1.05 (0.79, 1.38) | 1.01 (0.76, 1.34) | 1.02 (0.76, 1.36) |
| Jan 1, 1987 – Dec 31, 1999 | 11/11 | 97/84 | 1.04 (0.48, 2.25) | 0.92 (0.42, 2.01) | 1.07 (0.48, 2.36) |
| Jan 1, 2000 – Dec 31, 2009 | 25/19 | 119/106 | 1.06 (0.63, 1.80) | 1.14 (0.66, 1.95) | 1.08 (0.62, 1.87) |
| Jan 1, 2010 – Dec 31, 2018 | 37/42 | 119/140 | 1.02 (0.67, 1.54) | 0.92 (0.60, 1.43) | 0.96 (0.61, 1.47) |
| **Beyond 6 Months** | | | | | |
| All eras | 54/583 | 264/3,836 | 1.57 (1.12, 2.18) | 1.57 (1.11, 2.21) | 1.45 (1.03, 2.04) |
| Jan 1, 1987 – Dec 31, 1999 | 19/106 | 115/1,448 | 2.39 (1.20, 4.75) | 2.68 (1.28, 5.59) | 2.04 (0.95, 4.38) |
| Jan 1, 2000 – Dec 31, 2009 | 20/249 | 95/1,698 | 1.93 (0.99, 3.77) | 2.25 (1.12, 4.51) | 2.24 (1.11, 4.47) |
| Jan 1, 2010 – Dec 31, 2018 | 15/228 | 54/689 | 0.78 (0.38, 1.57) | 0.74 (0.35, 1.53) | 0.74 (0.41, 1.33) |

*Model 1: Adjusted for age, sex, body mass index, smoking status, cardiovascular etiology, malignancy, diabetes, hospital status, functional status, prior VAD, ECMO, life support status, total ischemic time, systolic blood pressure, inotropic medication usage, wait time, transplant era, HLA-AB mismatch, HLA-DR mismatch, PRA, acute rejection, bilirubin, creatinine, prior stroke, and history of cerebrovascular disease

**Model 2: Adjusted for all covariates in Model 1 and education level and insurance type

†Model 3: Adjusted for all covariates in Model 2 and setting of post-transplant care

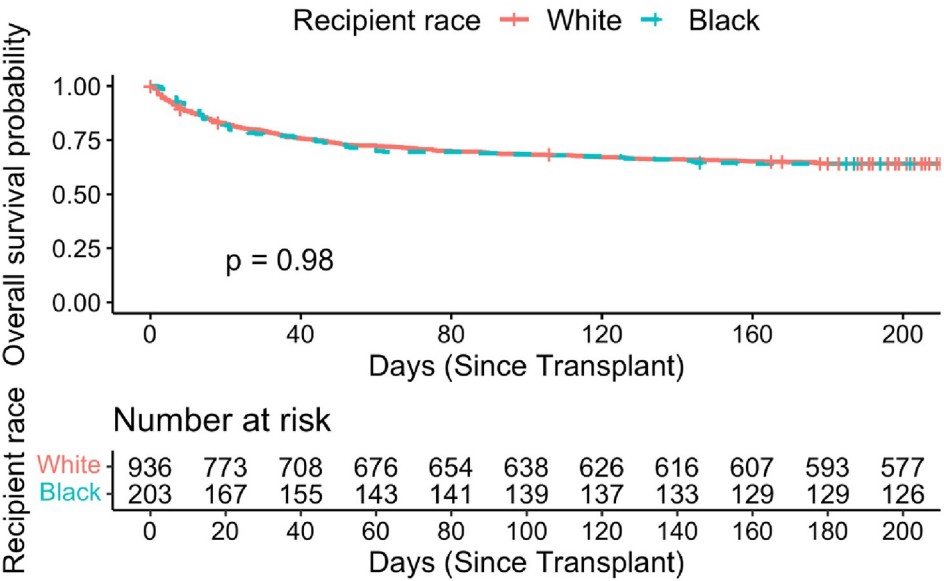

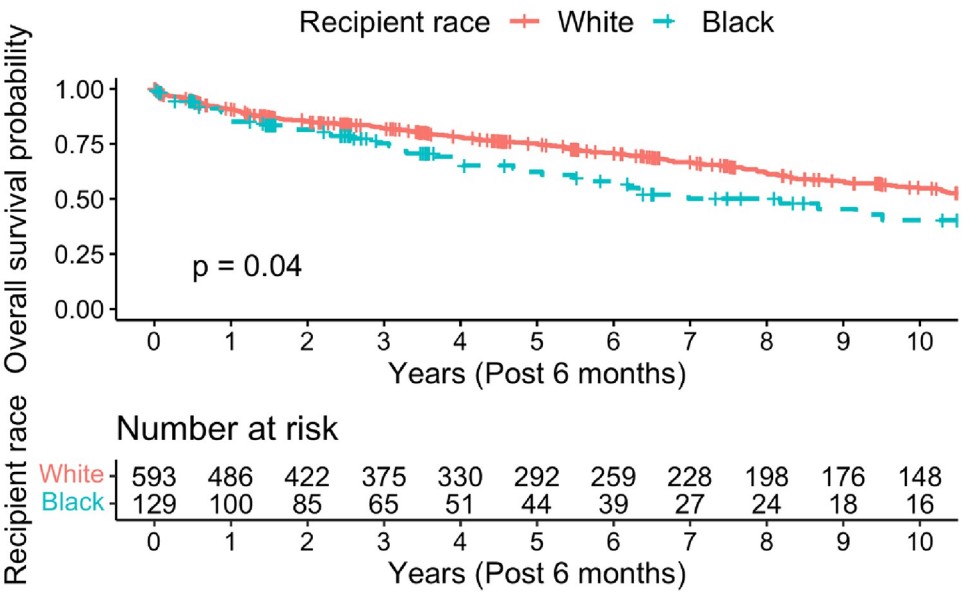

**Fig 1. Kaplan Meier survival curves for post-transplant stroke survivors by transplant era, Scientific Registry of Transplant Recipients (SRTR), January 1, 1987 To December 31, 2018.** (a) 1987–1999 (b) 2000–2009 (c) 2010–2018.

Black stroke survivors compared to white stroke survivors beyond 6 months following a stroke decreased over the years from 1987 to 2018 ($HR_{1987-1999}$ = 2.39, 95% CI: 1.20–4.75; $HR_{2000-2009}$ = 1.93, 95% CI: 0.99–3.77; $HR_{2010-2018}$ = 0.78, 95% CI: 0.38–1.57), with no apparent difference in the contemporary era ($HR_{2010-2018}$ = 0.78). Adjusting for education and insurance slightly strengthened the association in earlier eras ($HR_{1987-1999}$ = 2.68, $HR_{2000-2009}$ = 2.25), and slightly decreased the association in the most recent era ($HR_{2010-2018}$ = 0.74). After adjusting for the

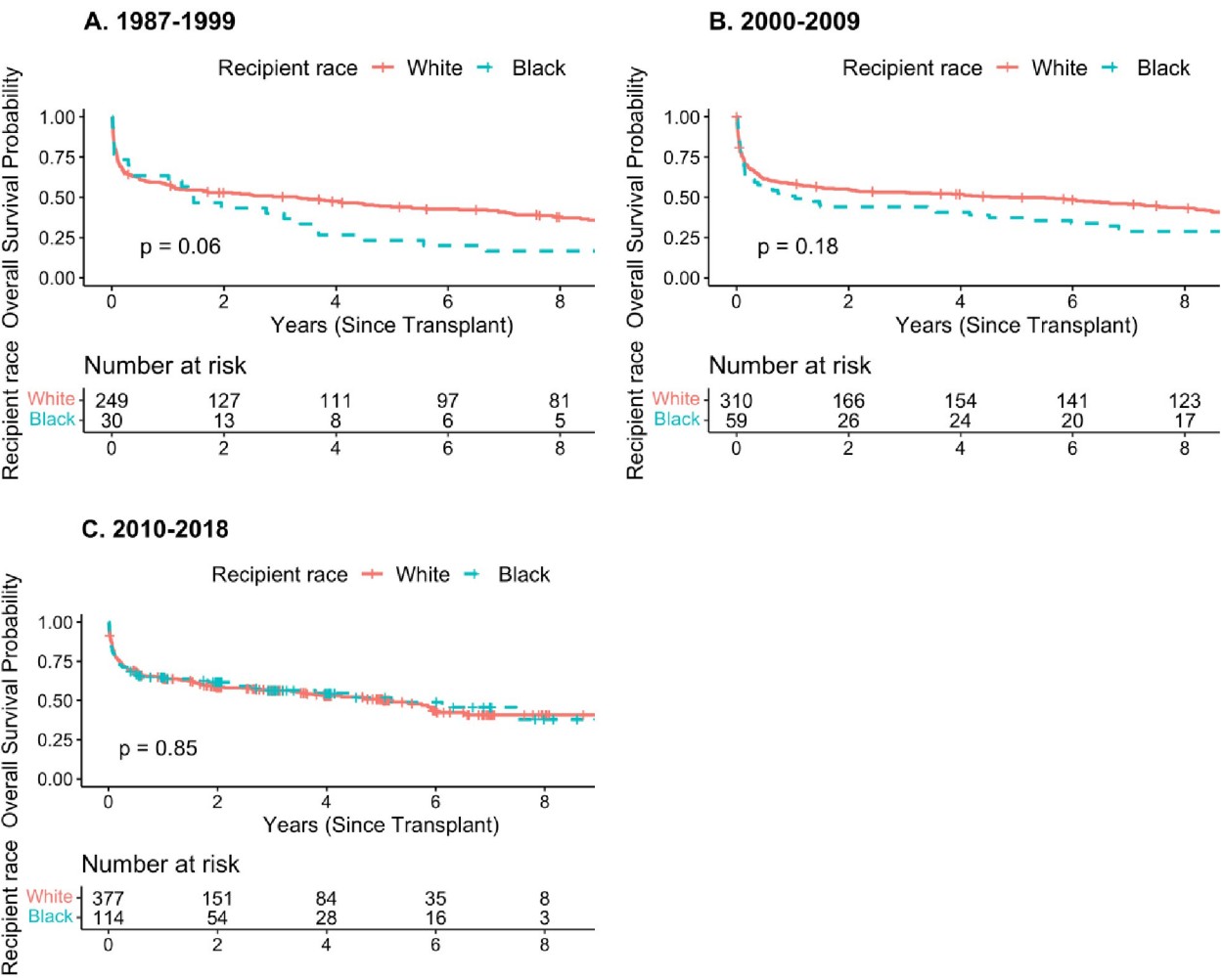

**Fig 2. Kaplan Meier survival curves for post-transplant stroke survivors, hazard ratios and 95% confidence intervals comparing mortality rates for Black post-transplant stroke survivors to mortality rates for white post-transplant stroke survivors according to era, Scientific Registry Of Transplant Recipients (SRTR), January 1, 1987 To December 31, 2018 (a) within 6 months of cardiac transplant and (b) beyond 6 months until last follow-up conditional on surviving beyond 6 months.**

post-transplant care setting, the adjusted hazard ratio for mortality was attenuated in the earliest transplant era ($HR_{1987-1999}$ = 2.04).

## Discussion

In this national registry study of adult heart transplant recipients, Black and white heart transplant recipients had a similar incidence of post-transplant stroke. Among cardiac transplant recipients who experienced a post-transplant stroke, there was a 22% higher rate of mortality among Black survivors than among white survivors over the full duration of follow-up, even after adjusting for clinical, demographic, and socioeconomic characteristics. Our findings are consistent with previous studies [6, 7, 11, 16, 17] reporting racial differences in mortality among heart transplant recipients. For instance, concordant with our findings among those with a post-transplant stroke, Maredia et al. found that between January 1, 2005, and January 31, 2017, Black recipients had a 29% (95% CI: 1.20–1.38) higher mortality rate than non-Black recipients. This disparity is greatest beyond 6 months after the stroke, and the association

appears to be mediated by differences in the post-transplant setting of care between Black and white patients. In the contemporary transplant era, Black patients had slightly better survival as white patients ($HR_{2010\text{-}2018}$ = 0.74, 95% CI: 0.41–1.33). This may reflect overall protocol improvements for heart transplant recipients irrespective of race, such as advancements in surgical techniques, immediate postoperative care, or differences in end-of-life care [18, 19]. We also note that the magnitude could be influenced by the extent of follow-up data we had, according to one study [20] reviewing the effect of amount of follow-up time on mortality hazard ratios; potentially if we had access to more recent follow-up data, we'd expect the association to tend towards 1.

When we examined the association between post-transplant stroke and mortality separately for *a priori* windows of the first six months following stroke and beyond 6 months after stroke, Black post-transplant stroke survivors had a higher mortality rate than white survivors (HR = 1.57, 95% CI: 1.12–2.18; Table 3, Model 1), but there was no association between race and mortality (HR = 1.05, 95% CI: 0.79–1.38). Further adjusting for socioeconomic variables such as education and insurance did not attenuate these associations between Black race and mortality within 6 months (HR = 1.01, 95% CI: 0.76–1.34) or beyond 6 months (HR = 1.57, 95% CI: 1.11–2.21). Several other studies [6, 7, 11] that used the same transplant registry and adjusted for education level and insurance type also did not report an attenuation of the association between Black race and mortality. There was no other socioeconomic information available in the SRTR registry, but we aimed to use such information to examine its role as a potential mediator rather than to explain the extent of racial differences by all components of socioeconomic status occurring later in life [21]. Although immunologic differences have been suggested as an explanation for the disparities [17], adjusting for immunologic factors did not attenuate the higher mortality rate experienced by Black patients. This is consistent with findings from Maredia et al. [6].

When we adjusted for the setting of long-term follow-up care, the association between race and mortality in post-transplant stroke survivors beyond 6 months was slightly attenuated. This suggests that differences in the setting of care for long-term follow-up may be at least partially responsible for the observed disparities. A study by Kilic et al. showed that a substantial proportion (56% vs. 47% of white patients) of Black patients underwent transplant procedures at lower performing centers, defined by higher observed-to-expected mortality ratios. We performed a sensitivity analysis stratifying by transplant center, which enables flexible baseline hazard estimation per transplant center and offers control over potential unmeasured confounders related to center-specific care practices and geographical variability. While this did not materially change our results, further data collection is needed to sufficiently power deeper analyses into center-specific care practice differences. Also, a higher proportion of deaths were attributed to lower treatment adherence in Black patients compared with white patients (1.% vs. 0.5%) [11]. Future studies should investigate factors that may be related to improvement in systems-level access to post-transplant care, education, and other resources to improve long-term mortality outcomes in Black post-transplant stroke survivors. Specifically, further investigating the setting of care as a potentially modifiable mediating factor may suggest that changing the setting of follow-up care, advocating closer co-management with specialists, or enhancing continuing education among healthcare professionals part of the transplant care team could improve survival.

Lastly, the mortality rate beyond six months after stroke among Black patients compared to white patients was higher over the total duration of the study, but it is driven by racial differences in early years and there is no difference in the recent era. Several single-center studies have suggested that advances in transplant recipient selection, care of patients awaiting a heart transplant, and post-transplant care of transplant recipients have contributed to early survival

improvements following transplantation for both white and Black patients [22–24]. A recent study using the UNOS database found that the historically poor survival of adult Black heart transplant recipients has gradually improved and became comparable to that of white transplant recipients [25]. General improvement in care, better compliance with the immunosuppression, reliable insurance coverage and increased awareness about reducing racial disparities in recent years could be contributing factors in improved survival of Black patients [26]. Another recent study using the UNOS database found that the one-year post-transplant survival for all adult transplant recipients improved in 2008–2017 compared to 2003–2007 despite the increased use of hearts from donors with risk factors as defined by cocaine use, intravenous drug use, hepatitis C virus positive, and opioid overdose death.

Our study has several limitations. The definition of racial categories in the SRTR transplant registry is, at best, a proxy for system factors that result in health care disparities [2, 27]. The number of Black post-transplant stroke survivors resulted in limited statistical power to examine disparities in mortality following post-transplant stroke in subsets of the population. However, our findings are consistent with previous literature. To our knowledge, this is the largest study of cardiac transplant recipients who experienced a post-transplant stroke. Although there is some missing data on covariates, the SRTR registry includes detailed information on several clinical and demographic factors. For some of the variables with a high proportion of missingness, we observed that the missingness was coming from the 1987–1999 transplantation era. However, the results and conclusions do not change even after running a sensitivity analysis in which we restricted the cohort to 2000–2018. Furthermore, keeping the cohort timeframe between 1987 and 2018 enables us to observe broader patterns in how racial differences in post-transplant mortality outcomes for post-transplant stroke survivors have changed over time. Although some misclassification may have occurred, our analyses of all-cause mortality are based on reliable outcome data from the Social Security Death Master File that has been linked to the SRTR transplant registry. We did not have detailed information on the clinical presentation and acute management of stroke nor NIHSS as SRTR is not strictly a stroke cohort dataset, but rather an administrative transplant dataset. However, SRTR remains a high-quality valuable dataset providing information on a wide set of clinical and demographic confounders, allowing for adjustment for differences in baseline characteristics in our multivariable-adjusted models [8].

In conclusion, we found that historically, the racial disparity in mortality is strongest in the period beyond the first 6 months and appears to have been mediated by differences between Black and white patients in the setting of long-term follow-up care. In the contemporary transplant era, the racial gap in mortality appears to have disappeared.

## Author Contributions

**Conceptualization:** Lathan Liou, Elizabeth Mostofsky, Murray A. Mittleman.

**Formal analysis:** Lathan Liou.

**Investigation:** Lathan Liou.

**Methodology:** Lathan Liou.

**Software:** Lathan Liou.

**Supervision:** Elizabeth Mostofsky, Murray A. Mittleman.

**Visualization:** Lathan Liou.

**Writing – original draft:** Lathan Liou.

**Writing – review & editing:** Lathan Liou, Elizabeth Mostofsky, Laura Lehman, Soziema Salia, Suruchi Gupta, Francisco J. Barrera, Murray A. Mittleman.

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
