## [Decision Letter · Decision Letter 0]

27 Jun 2022

PONE-D-22-12171 Racial Disparities in Post-Transplant Stroke and Mortality Following Stroke in Adult Cardiac Transplant Recipients in the United States PLOS ONE

Dear Dr. Liou,

Thank you for submitting your manuscript to PLOS ONE. After careful consideration, we feel that it has merit but does not fully meet PLOS ONE’s publication criteria as it currently stands. Therefore, we invite you to submit a revised version of the manuscript that addresses the points raised during the review process.

Please address the reviewer's comments. 

We look forward to receiving your revised manuscript.

Kind regards,

Nishant Kumar Mishra, MBBS PhD MD

Academic Editor

PLOS ONE

Journal Requirements:

 "EM is funded by Grant # K01AA027831 from the National Institute on Alcohol Abuse and Alcoholism (NIAAA). The content is solely the responsibility of the authors and does not necessarily represent the official views of the funding organizations."

Reviewers' comments:

Reviewer's Responses to Questions

**Comments to the Author**

1. Is the manuscript technically sound, and do the data support the conclusions?

Reviewer #1: Partly

Reviewer #2: Partly

2. Has the statistical analysis been performed appropriately and rigorously? 

Reviewer #1: I Don't Know

Reviewer #2: No

3. Have the authors made all data underlying the findings in their manuscript fully available?

Reviewer #1: No

Reviewer #2: Yes

4. Is the manuscript presented in an intelligible fashion and written in standard English?

Reviewer #1: Yes

Reviewer #2: Yes

5. Review Comments to the Author

Reviewer #1: In page 6 the authors state: “We defined post-transplant stroke as a stroke…... We adjusted for clinical and demographic factors identified by previous studies examining the association between race and mortality.” Then they go on to list 20 variables that they adjust for. In stroke cohorts baseline NIHSS and age were the major determinants of outcomes (See Uchino, Billheimer, Cramer Stroke 2001; Mandava, Kent 2009). Baseline NIHSS was the sole determinant of outcome for individual patients as well (Adams et al, Neurology 1999). Just because there were 900 patients having 20 variables does not give a reader added confidence in the study.

Registry data sets are notoriously unreliable (Willis et al, International Journal for Quality in Health Care, 2011), accentuating the already uncertain nature of heterogenous disease such as stroke.

Now adding imputation does not give the reader confidence regarding generalizability of the data. Garg in Acta Neurologica Scandinavica, 2022 has suggested that fragility index be used to assess the value of imputation.

That there are geographical disparities of stroke care in the US is well known. And that the authors cited Howard et al, Annals of Neurology 2011 is noted. However, there are other studies pointing to race based differential nature of response to thrombolytic treated of patients suffering a stroke (Mandava et al, Stroke 2013). However, a study published the same year (Mishra et al, International Journal of Stroke, 2013) did not find differences in response to thrombolytic treatment.

Xian et al, Annals of Internal Medicine 2011 report that blacks had lower mortality compared to whites. Xian et al and many others have discussed end of life care of black patients compared to white patients is different.

Would the authors attempt to explain the results of worse mortality in black heart transplant patients. Could this be due to geographic difference in care provided to blacks, response to treatment of black patients or choice of end of life care meted to blacks.

Reviewer #2: Thanks for the opportunity to review this manuscript. My concerns and suggestions are below.

1. Were patients from other racial groups (other than Black and White) included? If not a description of how many patients excluded and why will be useful. Also, when reviewing table 1 some numbers seem inaccurate; it is evident that transplant volume has grown in most recent period therefore, period of 2010 to 2018 (9 years) should have similar number of transplants performed compared to 1987-1999 (12 year period). In the table 1987-99 has more transplant performed which should carefully re-analyzed.

2. What was timing/definition of post-transplant stroke? Was this immediately post-transplant (within 30 days) or early (within 1-year)? This is very critical information. Must be elaborated.

3. Did the authors adjust for PA pressures, renal function (creatinine), Bilirubin, inotrope support?

4. How were PRA handled? Multiple values available. Were most recent ones used?

5. Based on your work it is evident that some variables have extensive missing values. Did you evaluate if these values were missing for a specific time frame? I.e. missing for earlier patients (prior to 1999). This may help you to limit the cohort to certain time period where you do not have to deal with extensive missing values. Also, consider the fact that most transplant have occurred in last 2 decades with more reliable and complete data available.

6. Did you consider pre-transplant stroke or history of cerebrovascular disease as a risk factor? It would be a valuable edition.

7. Although authors described the survival based on 3 time periods, they have not provided differences in transplant population between these time periods. It would be important to know population risk changes over time and its impact on survival outcomes.

6. PLOS authors have the option to publish the peer review history of their article (what does this mean?). If published, this will include your full peer review and any attached files.

Reviewer #1: No

Reviewer #2: **Yes: **Jaimin R. Trivedi

---

## [Author Response · Author response to Decision Letter 0]

25 Aug 2022

Please see the attached response letter for a detailed point-by-point response.

---

## [Decision Letter · Decision Letter 1]

14 Dec 2022

Racial Disparities in Post-Transplant Stroke and Mortality Following Stroke in Adult Cardiac Transplant Recipients in the United States

PONE-D-22-12171R1

Dear Dr. Liou,

We’re pleased to inform you that your manuscript has been judged scientifically suitable for publication and will be formally accepted for publication once it meets all outstanding technical requirements.

Kind regards,

Nishant Kumar Mishra, MBBS PhD MD

Academic Editor

PLOS ONE

Additional Editor Comments (optional):

Reviewers' comments:

Reviewer's Responses to Questions

**Comments to the Author**

1. If the authors have adequately addressed your comments raised in a previous round of review and you feel that this manuscript is now acceptable for publication, you may indicate that here to bypass the “Comments to the Author” section, enter your conflict of interest statement in the “Confidential to Editor” section, and submit your "Accept" recommendation.

Reviewer #2: All comments have been addressed

2. Is the manuscript technically sound, and do the data support the conclusions?

Reviewer #2: Yes

3. Has the statistical analysis been performed appropriately and rigorously? 

Reviewer #2: Yes

4. Have the authors made all data underlying the findings in their manuscript fully available?

Reviewer #2: Yes

5. Is the manuscript presented in an intelligible fashion and written in standard English?

Reviewer #2: Yes

6. Review Comments to the Author

Reviewer #2: All comments addressed. No additional comments. Manuscript is now acceptable.

7. PLOS authors have the option to publish the peer review history of their article (what does this mean?). If published, this will include your full peer review and any attached files.

Reviewer #2: **Yes: **Jaimin Trivedi

---

## [Editor Report · Acceptance letter]

20 Dec 2022

PONE-D-22-12171R1 

Racial disparities in post-transplant stroke and mortality following stroke in adult cardiac transplant recipients in the United States 

Dear Dr. Liou:

I'm pleased to inform you that your manuscript has been deemed suitable for publication in PLOS ONE. Congratulations! Your manuscript is now with our production department. 

Kind regards, 

on behalf of

Dr. Nishant Kumar Mishra 

Academic Editor

PLOS ONE